# Liquid Biopsy in the Oncological Management of a Histologically Undiagnosed Lung Carcinoma: A Case Report

**DOI:** 10.3390/jpm12111874

**Published:** 2022-11-09

**Authors:** Giovanni M. Fadda, Renato Lobrano, Milena Casula, Marina Pisano, Antonio Pazzola, Antonio Cossu, Giuseppe Palmieri, Panagiotis Paliogiannis

**Affiliations:** 1Medical Oncology Unit, University Hospital (AOU) of Sassari, Via Enrico De Nicola 1, 07100 Sassari, Italy; 2Anatomic Pathology and Histology, University Hospital (AOU) of Sassari, Via Matteotti 60, 07100 Sassari, Italy; 3Institute of Genetic and Biomolecular Research, National Research Council (CNR), Traversa La Crucca 3, 07100 Sassari, Italy; 4Department of Medicine, Surgery and Pharmacy, University of Sassari, Viale San Pietro 43, 07100 Sassari, Italy; 5Department of Biomedical Sciences, University of Sassari, Viale San Pietro 43, 07100 Sassari, Italy

**Keywords:** lung cancer, adenocarcinoma, targeted therapy, EGFR, liquid biopsy

## Abstract

Lung cancer is one of the most common and lethal cancers worldwide. Numerous medications targeting specific molecular alterations in non-small cell lung cancer have been introduced in the last decade and have revolutionized the clinical management of the disease. Their use has brought to a parallel evolution of molecular testing techniques to identify alterations in druggable molecular targets within the genetic material of the tumors. To perform molecular testing, biopsy or surgery tissue specimens are needed, which in addition allow the histological characterization of the tumors. Unfortunately, in real-life practice not all the patients are suitable for biopsy or surgery procedures. The use of liquid biopsy for blood extracted tumoral DNA analysis is a promising approach in unbiopsied cases, but it is also weighted by several methodological and technical limitations. We report here a case of histologically undiagnosed lung cancer managed with a liquid biopsy and subsequently with anti-EGFR treatment. Our report highlights that the use of liquid biopsy molecular testing in specific clinical situations can offer treatment opportunities for fragile patients affected by lung cancer.

## 1. Introduction

Lung cancer is one of the most common and lethal malignancies worldwide [1,2]. In 2020, the Global Cancer Observatory estimated approximately 2,207,000 new cases and 1,796,000 deaths for lung cancer [1]. Incidence rates have progressively decreased in males and increased in females in the last three decades, mainly due to changes in tobacco smoking habits between genders [2,3]. Tobacco smoking is the most relevant causal factor of lung cancer, often occurring together with other risk factors like radon exposure in indoor environments and mines, occupational factors (i.e., asbestos), air pollution and other risk factors [4]. Lung cancer is traditionally divided into two large subgroups, small-cell lung cancer (SCLC) and non-small cell lung cancer (NSCLC). The latter encompasses several histologic subtypes, including lung adenocarcinoma, squamous cell carcinoma (SCC), large cell carcinoma and other histologic subtypes. In the USA, SCLC was the most common subtype in the late 1980s, SCC in the early 1990s and lung adenocarcinoma in the late 1990s, reflecting differences in the construction of smoking cigarettes, as well as in the composition of tobacco and inhalation patterns [5].

High mortality rates, relatively close to those of incidence, are still observed in lung cancer. This depends mostly on the fact that the vast majority of patients (80–90%) are asymptomatic at the time of the initial diagnosis [2]. Signs and symptoms of the disease, related to the primary tumor (cough, hemoptysis and dyspnea, etc.), or to metastases (bony pain, weight loss and fatigue, etc.), along with laboratory test alterations, become clinically evident in locally advanced or metastatic stages. In these stages, systemic oncological treatments are necessary, often in combination with local treatments. Chemotherapy has been the mainstem of lung cancer medical treatment for several decades; unfortunately, with poor survival outcomes and significant toxicities.

Consistent improvements in the clinical management and oncological outcomes of patients with NSCLC have been achieved in the last decade, with the introduction of novel treatment strategies such as immunotherapy and medications targeting specific gene alterations [6]. In particular, a number of drugs targeting alterations of the epidermal growth factor receptor (EGFR), Kirsten rat sarcoma (KRAS), v-Raf murine sarcoma viral oncogene homolog B (BRAF), anaplastic lymphoma kinase (ALK), protooncogene C-Ros-1 (ROS1), hepatocyte growth factor receptor (MET), rearranging during transfection (RET) and neurotrophic tyrosine receptor kinase (NTRK1/2/3) genes have been recently approved for patients affected by advanced stage NSCLC [6,7,8]. A variable, but consistent proportion of patients with lung adenocarcinoma, harbor at least one of these genetic alterations, with a prevalence depending on race, age and other factors. EGFR mutations on exons 18, 19 and 21 are the most commonly observed alterations, occurring in 10–15% of Caucasians and up to 50% or more in Asian patients with lung adenocarcinoma [6]. Several methods and technologies have been developed for molecular testing, in either tumoral tissues or peripheral blood [9]. Molecular testing is generally performed in tissue samples obtained through tumor biopsies or surgery, for diagnostic purposes. Indeed, a histological diagnosis of NSCLC and the specific histotype need to be established prior to clinical staging and molecular profiling. Nevertheless, in some severe clinical cases no tissue samples are obtainable for histological evaluation and molecular testing, neither by biopsy nor surgery, thus the oncological management of the patients is particularly difficult [10,11,12]. We describe herein, a case of an old patient with a highly suspected, but histologically undiagnosed advanced NSCLC, who successfully underwent anti-EGFR-targeted therapy prescribed after a liquid biopsy.

## 2. Case Report

An 87-year-old female patient was referred to the Emergency Unit of our hospital for severe dyspnea, coughing, weakness and malaise in October 2021. In the patient’s past medical history, the main issues were a myocardial infarction treated in 2012 with percutaneous transluminal coronary angioplasty and coronary stenting, as well as carotid atherosclerosis, hypertension, asthmatic bronchitis, osteoporosis and severe hearing loss; no family history for lung cancer was reported. On physical examination, signs of a pleural effusion with lung atelectasis were evidenced and an emergency thoracentesis was performed. Dyspnea was only slightly improved, thus a computed tomography (CT) scan of the thorax was done, showing a pneumothorax and a residual pleural effusion, along with a large pulmonary mass involving the upper lobe of the right lung, several smaller lesions in the remaining parenchyma and enlarged ipsilateral hilar and mediastinal lymph nodes (Figure 1A,C). A thoracic tube was subsequently placed to treat both the pneumo- and hydrothorax. Unfortunately, no pleural liquid sampling was performed for pathological testing at this stage, and the poor clinical condition of the patient did not allow any endoscopic or imaging-guided biopsy to pathologically establish the nature of the disease.

The case was discussed in the thoracic multidisciplinary team of our hospital; the difficulties in performing a tissue biopsy, the high probability of an NSCLC, the fact that the patient had never been a smoker, and the relevant global load of the disease led to the decision to perform a liquid biopsy to search for EGFR mutations in the circulating tumoral DNA (ctDNA). The patient was adequately informed and gave her written consent for both the molecular testing and the publication of her anonymous clinical data. Thirty milliliters of venous blood was collected with standard venipuncture procedures and sent for testing in December 2021. For the ctDNA isolation from plasma, the fresh peripheral blood samples collected in EDTA-containing tubes were firstly centrifuged at 1900× *g* for 10 min to plasma separation, then further centrifuged at 16,000× *g* for 10 min for removing any residual debris. About 4 mL of plasma was finally collected; an aliquot of 2 mL was used to isolate the ctDNA for the Therascreen assay and the remaining 2 mL aliquot was processed on the Idylla System for result confirmation. The ctDNA was extracted using the QIAamp circulating nucleic acid kit on the QIAVac 24 Plus connected to a vacuum pump, according to the manufacturer’s instructions. The concentration of the purified ctDNA was assessed by using the Qubit 2.0 Fluorometer and the Qubit dsDNA HS (high sensitivity) assay kit (Life Technologies, Carlsbad, CA, USA).

Molecular testing was performed using the Therascreen EGFR Plasma RGQ PCR kit (Qiagen GmbH, Hilden, Germany), which was used as a recognized reference method following the manufacturer’s instructions. In particular, analysis is based on a real-time PCR assay that combines an amplification refractory mutation system (ARMS) and a Scorpion fluorescent primer/probe system. The allele-specific amplification is achieved by the ARMS, which exploits the ability of Taq DNA polymerase to distinguish between a matched and a mismatched base at the 3′ end of a PCR primer. When the primer is fully matched, the amplification proceeds with full efficiency. When the 3′ base is mismatched, only low-level background amplification may occur. Therefore, a mutated sequence is selectively amplified even in samples where most of the DNA does not carry the mutation. Detection of amplification is performed using Scorpions, which are bifunctional molecules containing a PCR primer covalently linked to a probe. The probe incorporates the fluorophore, carboxyfluorescein (FAM), and a quencher turning off the fluorescence of the fluorophore. When the probe binds to the ARMS amplicon during PCR, the fluorophore and quencher become separated, leading to a detectable increase in fluorescence. The Therascreen EGFR RGQ PCR kit enables the detection of the following mutations: the main TKI-sensitive deletions on exon 19, the TKI-resistant T790M mutation on exon 20 and the TKI-sensitive L858R mutation on exon 21, with no chance to distinguish the different deleted alleles into the multiplexed array. The Rotor-Gene Q MDx instrument was used to perform the real-time qualitative PCR assay for the detection of somatic mutations in the *EGFR* gene, using genomic DNA extracted from the liquid biopsy samples.

In our case, the Therascreen EGFR RGQ assay detected the existence of an EGFR exon 19 deletion. In Appendix A, the graphical representation of the results of our mutation analysis is reported. To confirm the result, we processed the second aliquot of 2 mL of plasma on the Biocartis Idylla System, a fully automated liquid biopsy assay. The Idylla technology is cartridge-based and uses microfluidic processing (capillary action-based pumping) with all the reagents on-board. The cartridges are loaded with 2 mL of plasma and 20 µL of Proteinase K (20 mg/mL) and the remaining processes, including extraction and the real-time PCR of EGFR ctDNA and the data analysis, are completely automated. The technology used by Idylla combines the allele-specific primers’ amplification using PlexPrimers with allele-specific detection using PlexZymes (also known as MNAzymes) [13,14,15]. Each PlexPrimer contains an “insert sequence”, positioned between the 5′ and 3′ target-specific regions, which is non-complementary to the target initially, but is introduced into amplicons during amplification. For multiplexed mutation detection, each PlexPrimer, containing a different INS sequence, is designed to be allele-specific via the complementarity of the 3′ terminus of the target mutation. The Idylla ctEGFR Mutation Assay is designed for research purposes and covers 40 different *EGFR* variations: G719A/C/S in exon 18, 28 deletions in exon 19, T790M and S768I, as well as 5 insertions in exon 20, and L858R and L861Q in exon 21. The Idylla assay confirmed that the patient carried a circulating somatic EGFR exon 19 deletion.

Therefore, the patient started treatment with the oral administration of Osimertinib at 80 mg/day. No side effects or relevant toxicities occurred until the first follow-up evaluation, which was performed eight weeks later. The clinical condition of the patient was substantially improved, including the pleural effusion, dyspnea and cough, while a thoracic CT scan showed a consistent reduction of the pulmonary nodules (Figure 1A,B), along with the hilar and mediastinal lymph nodes (Figure 1C,D). After 12 weeks, the treatment continued regularly and was well-tolerated by the patient.

## 3. Discussion

Molecular testing for the use of gene-addicted therapies is essential in modern clinical practice, especially in patients with advanced NSCLC. Generally, molecular analyses in druggable genes are performed in biopsy or surgery tissue specimens. Biopsy specimens in patients with lung cancer are generally obtained with endoscopic or radiological image-guided methods characterized by a certain degree of invasiveness and risk for failure or complications [11]. In particular, failure to obtain appropriate tumoral tissue in terms of both quantity and quality, adverse events in image-guided techniques (~17%) or pneumothorax requiring chest tube placement (14%), represent the main limitations for performing lung tumor biopsies [16]. In our case, the clinical condition of the old patient was critical, due to the presence of severe dyspnea and a pleural effusion, which required an emergency thoracentesis that subsequently caused a pneumothorax and determined the need for drainage tube placement. In similar cases the need for alternative biological matrices for molecular testing is urgent, to avoid exclusion of a consistent amount of patients from targeted therapies.

Liquid biopsy of peripheral blood and molecular testing in circulating DNA was recently introduced in clinical practice, mainly for targeted therapy monitoring and detection of arising molecular alterations that cause resistance to drugs; it has the advantage of being minimally invasive, repeatable, and offering a global landscape of the molecular profile of the tumor, beyond its clonal heterogeneity [17,18]. On the other hand, it is limited by the amount and duration of the ctDNA, the costs and the technical issues regarding both the collection of the samples and the sensitivity and specificity of the sequencing methods adopted [19,20]. Initial trials performed on serum showed a low sensitivity of liquid biopsies in detecting EGFR mutations in comparison to tissue testing, but subsequently the introduction of plasma testing and the guidelines for sample collection and storage, along with the introduction of better sequencing technologies, improved the performance of the blood biopsy and stimulated research for the use of other body fluids for molecular testing (saliva and urine, etc.) [21]. Nevertheless, the use of a liquid biopsy in histologically undiagnosed NSCLC is only anecdotal, and its usefulness remains to be investigated. We successfully adopted this approach in our case, as the clinical and oncological improvement of our patient was significant; furthermore, this made feasible a future tissue biopsy for comparative molecular testing with liquid biopsies, in accordance with our patient’s needs.

EGFR mutations were the first discovered druggable genetic alterations in NSCLC. EGFR is a transmembrane protein with multiple functions in normal cell survival and growth. It has been observed that EGFR was often overexpressed and aberrantly activated in NSCLC, with several activating mutations within the kinase domain of the EGFR gene, particularly in lung adenocarcinomas [22,23]. Consequently, these tumors were highly sensitive to EGFR-tyrosine kinase inhibitors (TKIs). TKIs were progressively adopted in clinical practice, offering an important therapeutic option in patients with lung adenocarcinomas. However, the increased frequency of resistance to first and second generation TKIs reduced the initial enthusiasm associated with the use of these therapeutic agents [24]. Currently, third generation TKIs like Osimertinib are available, with fewer resistances and thus, better survival outcomes. Resistance to first and second generation TKIs was mainly caused by the occurrence of a T790M EGFR mutation in exon 20; this mutation was not detected in our case. Our patient had an EGFR exon 19 deletion and was treated successfully with Osimertinib. Molecular alterations in EGFR exon 19 are the most common druggable targets in lung adenocarcinoma, amounting for about 45% of total genetic alterations; EGFR exon 20 (40–45%) and EGFR exon 18 (5%) represent further therapeutic targets, while EGFR exon 20 targets (approximately 1%) are rarer and often lead to treatment resistance [22]. Mutations which confer resistance to TKIs may arise in the primary tumor, or even subsequently during treatment, under the evolutive pressure of the TKIs themselves; liquid biopsy can play a double role in this context, allowing both the initial testing for EGFR mutations and the subsequent follow-up and search for genetic alterations leading to treatment resistance and disease progression.

In addition, as technological improvements progress and the sensitivity and specificity of NGS techniques increase, a wider spectrum of molecular alterations can be detected. In particular, the V600E mutation in the BRAF gene and the G12C mutation in the KRAS gene can be searched, both of which are druggable in NSCLC [25]. Similarly, gene fusions in ALK, ROS1, MET, RET and NTRK genes can be detected through RNA-based NGS techniques, in order to add further curative options for patients with NSCLC. In addition, higher levels of sensitivity and specificity rates can be reached, in comparison with the current reference tests. Garcia et al. recently reported a sensitivity of 98.5% and a specificity of 98.9% using NGS cfDNA sequencing via molecular amplification pools (MAPs), with a digital droplet PCR assay as the reference, and detection of somatic variants in 73% of 356 lung cancer patients receiving plasma testing as part of routine clinical management [26]. These complex and expensive technologies are not widely available currently; our experience showed that simpler and less expensive instruments like the Therascreen PCR Assay are extremely useful in clinical practice when NGS platforms are not available. 

## 4. Conclusions

NSCLC is one of the most common and lethal cancers worldwide. The introduction of targeted therapies for its treatment in the last decade have caused an improvement in the survival of patients with NSCLC, and have made molecular testing compulsory in order to select the appropriate medications and therapeutic strategies. Molecular testing is generally performed in tissue specimens. Nevertheless, liquid biopsy represents a valid tool for molecular testing and an option that is useful for clinical decisions in selected patients with histologically undiagnosed, yet highly suspected NSCLC, for which tissue samples cannot be obtained. Druggable molecular alterations in circulating tumoral genetic material can be detected this way, allowing the use of targeted medications to treat patients in an advanced stage of the disease, including those with poor clinical conditions. The introduction of NGS molecular testing technologies will expand the ability to detect molecular genetic alterations through liquid biopsy, both in the initial phase and during the treatment of patients with NSCLC.

## Figures and Tables

**Figure 1 jpm-12-01874-f001:**
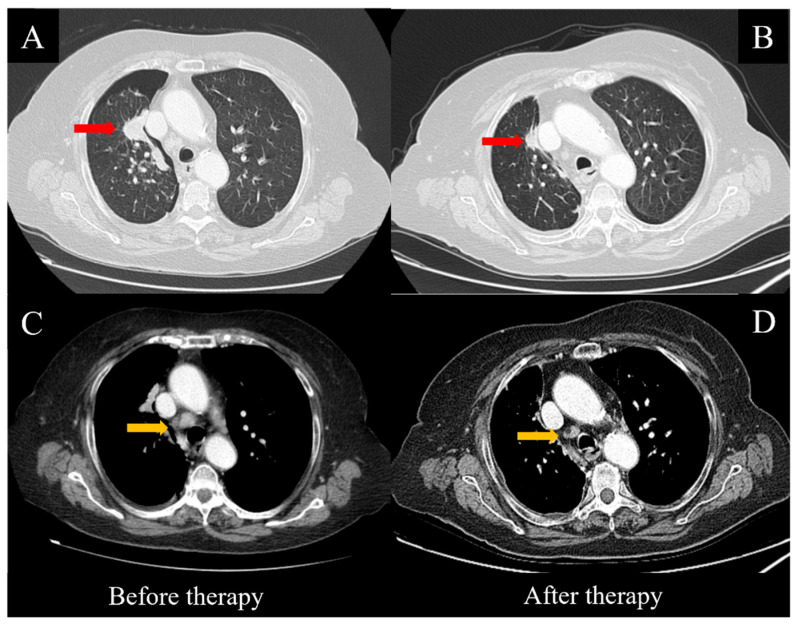
CT scans before and after targeted therapy with Osimertinib. Consistent regression of the main mass (red arrows, **A**,**B**) and the mediastinal lymph nodes (yellow arrows, **C**,**D**) was observed after eight weeks of treatment.

## Data Availability

Not applicable.

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
