# Peer review of "Liquid Biopsy in the Oncological Management of a Histologically Undiagnosed Lung Carcinoma: A Case Report"

_jpm, 2022, doi:10.3390/jpm12111874_

Round 1

Reviewer 1 Report

This article described an unusual situation to apply liquid biopsy in determining the use of EGFK TKI in an undiagnosed lung cancer patient, and achieved a successful outcome. This article may be interesting to readers, but some more information should be provided.

Some concerns are as follows:

1.     The authors should briefly describe how they performed the liquid biopsy, for example, the blood amount or tubing they used. Did they repeat the testing?

2.     Most importantly, the result of liquid biopsy should be well shown to reveal the detection of EGFR exon 19 deletion.

3.     What did the “T” mean in the first sentence of discussion?

Author Response

Dear reviewer

Please find enclosed the revised version of our manuscript. We would like to thank you for your valuable suggestions. All revisions have been made based on your suggestions, are highlighted in yellow in the text and listed here point by point.

Reviewer 1

Issue 1. The authors should briefly describe how they performed the liquid biopsy, for example, the blood amount or tubing they used. Did they repeat the testing?

Reply: Some additional details regarding blood sampling have been added in the text. The test was not repeated.

Issue 2. Most importantly, the result of liquid biopsy should be well shown to reveal the detection of EGFR exon 19 deletion.

Reply: The result of the test was better described, and a figure illustrating it has been added as Supplementary Figure 1.  

Issue 3. What did the “T” mean in the first sentence of discussion?

Reply: “T” in the first sentence of discussion was just a typing error that has now been corrected.

We look forward to hearing from you.

Best regards

Reviewer 2 Report

In this case report, the authors described a clinical course of a 87 years old female patient with a possibility of lung cancer. The liquid biopsy with the Therascreen EGFR Plasma RGQ PCR kit found an EGFR exon 19 deletion mutation, which led the treatment with osimertinib.

1. Because thoracentesis was performed for this patient, the reviewer thinks that pleural effusion should be analyzed pathologically using cell block technique. Then, the uncertain liquid biopsy was not necessary for this patient.

2. This report just describes the clinical course of "a lucky patient", because the possibility for the successful detection of the EGFR mutation will be less than 50% (she may had other driver mutation or her tumor may not shed ctDNA). Therefore, it cannot be said that the content is constructive.

Author Response

Dear reviewer

Please find enclosed the revised version of our manuscript. We would like to thank you for your valuable suggestions. All revisions are highlighted in yellow in the text, and listed here point by point.

Issue 1. Because thoracentesis was performed for this patient, the reviewer thinks that pleural effusion should be analyzed pathologically using cell block technique. Then, the uncertain liquid biopsy was not necessary for this patient.

Reply: Thoracentesis has been performed in an emergency setting, and unfortunately non cell-block or other sampling has been made until the thoracic tube has been removed. This is the reason we subsequently choose to perform a liquid biopsy. We added a phrase in the text to make it clearer. We agree with the reviewer that pleural effusion sampling should be always performed, especially in critical cases like ours.  

Issue 2: This report just describes the clinical course of "a lucky patient", because the possibility for the successful detection of the EGFR mutation will be less than 50% (she may had other driver mutation or her tumor may not shed ctDNA). Therefore, it cannot be said that the content is constructive.

Reply: It is true that the patient was a “lucky” patient, but we would never know it if we hadn’t performed the testing. This is the point of our report, we just describe an option which should be kept in mind by clinicians managing similar cases, considering that molecular analysis is performed in just a blood sample, which may avoid more invasive options. We added a brief phrase in the conclusions to underline it.

We look forward to hearing from you.

Best regards

Reviewer 3 Report

I have reviewed the case report by Giovanni M. Fadda et al. The authors have identified an EGFR mutation in a patient from a liquid biopsy. The patient probably had lung cancer but a biopsy from the tumor could not be performed. Overall, I think it is an interesting case. I have two minor points:

It would be informative with arrows on the scans, so the researcher who normally doesn’t assess, can interpret the data with scan imaging.

The deletion of exon 19 in EGFR is not described. The author should add a reference or show data that loss of exon 19 in EGFR results in constitutive activation and therefore suitable for drug treatment.

Author Response

Dear reviewer

Please find enclosed the revised version of our manuscript. We would like to thank you for your valuable suggestions. All revisions are highlighted in yellow in the text, and listed here point by point.

Issue 1. It would be informative with arrows on the scans, so the researcher who normally doesn’t assess, can interpret the data with scan imaging.

Reply: Colored arrows have been added in the images, and the corresponding explanations in the figure legend.

Issue 2. The deletion of exon 19 in EGFR is not described. The author should add a reference or show data that loss of exon 19 in EGFR results in constitutive activation and therefore suitable for drug treatment.

Reply: The result of the test was better described, and a figure illustrating it has been added as Supplementary Figure 1.

We look forward to hearing from you.

Best regards

Reviewer 4 Report

The submission by Paliogiannis et al. presented a case of a liquid biopsy followed by anti-EGFR treatment. Liquid biopsy indeed a useful technique for the case where surgery is not possible or an additional method to measure the mutation diagnosis and treatment follow up. The work is very well described; however, it requires few points to be mentioned before considering for publication, which are mentioned below. 

1)     This reviewer understand that EGFR is the most common mutation in NSCLC, however apart from these other mutations / fusions are also likely to be occurred. So why authors have just chosen EGFR screening or have they performed any additional mutations screening and not discussed here.  

2)     Since the report is focused on molecular screening followed by treatment and follow-up. So, does the authors have performed any molecular screening for disease marker analyzing disease progression or resistance occurrence at the time of follow up? If it is done, then please report the findings in graphical form as mentioned in other manuscript. 

3)     It will be good for the audience if authors can provide details about the molecular testing like brief scientific method and scope of such technique.  

4)     There are some editorial works required at line 45 and 93. Please update that. 

Author Response

The authors would like to thank the reviewer for his/her valuable suggestions and corrections.

  1. At the time of the case described we had the technologies and kits to only test EGFR mutations in liquid biopsy. It is absolutely true that there are other genetic alterations to investigate in patients with suspected or confirmed lung cancer, as describe in the introduction of our manuscript. We added in the “Discussion” a comment on the new technologies which more and more allow to detect a wider spectrum of molecular abnormalities.
  2. We did not perform any further testing, as the patient showed a significant clinical improvement with the current therapy. We had no resistances at the time of the last follow-up.  
  3. We added a detailed description of the molecular technique used in our case.
  4. Editorial errors were corrected

Round 2

Reviewer 1 Report

Congratulations ! The authors have made significant change of the manuscript and it is now acceptable for publication.

Author Response

The authors would like to thank the reviewer for his/her valuable suggestions. 

Reviewer 2 Report

The reviewer understands the reason why the authors could not perform genetic testing using pleural effusion.

Author Response

(The authors gave the same response as above.)

Reviewer 4 Report

Thank you for providing the detailed information about the method and other concerns. The reviewer thinks that brief method will be good to include in the main text. If author wants to include detailed report than supplementary section would be an ideal option. 

Please correct the dose to 80 mg/day instead of 80mg/die. 

Author Response

Dear reviewer

Thank you for your suggestions. Here is our reply point by point.

  1. We added a detailed description of the testing method because it was requested by the editor. The editor asked to extend the text of the paper to more than 2.500 words to meet their editorial criteria and policies. We did that by adding new paragraphs in “Introduction” and “Discussion”, as well as technical details regarding the molecular testing in the “Case Report” section. For this reason, we cannot reduce the text moving parts in “Supplementary material” where we inserted a graph of the genetic alteration found.
  2. We apologize for the error, that was not corrected in the last revision. Now we corrected it truly.

For and behalf of the Authors

Best regards